# Association between Social Engagements and Stigmatization of COVID-19 Infection among Community Population in Japan

**DOI:** 10.3390/ijerph19159050

**Published:** 2022-07-25

**Authors:** Yuna Koyama, Nobutoshi Nawa, Yui Yamaoka, Hisaaki Nishimura, Jin Kuramochi, Takeo Fujiwara

**Affiliations:** 1Department of Global Health Promotion, Tokyo Medical and Dental University (TMDU), Tokyo 113-8519, Japan; yui.yamaoka@gmail.com (Y.Y.); hisaaki.nishimura@gmail.com (H.N.); fujiwara.hlth@tmd.ac.jp (T.F.); 2Department of Medical Education Research and Development, Tokyo Medical and Dental University (TMDU), Tokyo 113-8519, Japan; nawa.ioe@tmd.ac.jp; 3Kuramochi Clinic Interpark, Tochigi 321-0114, Japan; raouyuria@gmail.com

**Keywords:** stigma, pandemic, COVID-19, social capital, social network, social punishment

## Abstract

In the face of unknown risks, including the coronavirus disease 2019 (COVID-19) pandemic, we tend to have stigmatized perceptions. The current study aimed to examine the association of social engagements with the level of stigmatization of COVID-19 infection among the general population. The data of 429 participants of the Utsunomiya COVID-19 seroprevalence neighborhood association (U-CORONA) study, a population-based cohort study conducted in Utsunomiya City, Japan, were analyzed. Their stigmatized perception of people with COVID-19 infection was evaluated via a questionnaire for the situation if they or others in their community were to get infected. The association between social engagements (community social capital, social network diversity, and social network size) and stigmatization were analyzed by a multiple linear regression model with generalized estimating equations. Overall, females reported a higher stigmatized perception of people with COVID-19 than males. Lower education and depressive symptoms were also positively associated with higher stigmatization, while age, household income, and comorbidities were not. People with higher community social capital reported lower stigmatization (B = −0.69, 95% CI = −1.23 to −0.16), while social network diversity and social network size did not show an association with stigmatization. We found an association between community social capital and stigmatization, suggesting that enhancing their community social capital, but not social network diversity and size, has the potential to mitigate the levels of stigmatization.

## 1. Introduction

Stigmatization is a process to differentiate and separate those who are culturally devalued and discredited from the whole society [1]. It was observed commonly in the face of the unknown and uncertainty [2,3] since it provides social status dominant over the out-groups [3], together with a clear explanation and cause of fear and danger [4]. As a consequence of being stigmatized, burnout syndrome [5], depression, and anxiety [6] are frequently observed. Furthermore, stigmatization could impact the allocation of resources and healthcare access [3,7,8], which may exacerbate the disparity. From the public health perspective, stigmatization would hinder rapid and effective policy intervention by increasing already existing social inequalities and impeding social integrations [2]. Therefore, knowing what the effective target point is to change stigmatized perceptions is urgent.

The coronavirus disease 2019 (COVID-19) pandemic, of which prevention, symptoms, and treatment remained much unknown in 2020 [9], put people at high risk of stigmatization. We observed stigmatization, especially toward healthcare workers and survivors of the previous pandemic of the Middle East respiratory syndrome (MERS) and severe acute respiratory syndrome (SARS), acquired immunodeficiency syndrome (AIDS), and Ebola [10,11,12,13]. Thus, understanding stigmatization during the COVID-19 pandemic would not only benefit the current situation, but also future pandemics.

Under the current COVID-19 pandemic, there have been extensive reports on the stigmatization of healthcare workers [3,6,13,14]. Minority groups, such as those who have a history of incarceration and psychiatric illness [7], older population [8], people of Asian ethnicity [3,14,15,16], and those who recovered from COVID-19 [17,18], are also the targets of stigmatization. In addition, people who survived the disease and who resided in high pandemic areas felt stigmatized derived from guilt and shame due to their affiliations and beliefs of being excluded [16,17,18]. Despite the growing evidence on the people at risk of being stigmatized, there has been little research on who is at risk of stigmatization among the general population, with stigmatization being insufficiently assessed with one or few questions [15,19,20].

Stigmatization lies at the interface of community and individual factors. That is, stigmatization occurs from both individual desire to separate themselves from the out-groups and community stereotypes of who is categorized as the out-group [21]. Those who are highly engaged in their community consider people in their community as in-groups and have a lower stigmatized perception, whereas those who are not engaged as much would have stronger stigmatization toward people in their community. As an indicator of their social engagements, we focused on the roles of social capital defined by trust, ties, and mutual aid with the community (as an emotional connection), and social network diversity and size (as a structural connection). Social engagements would provide potential target points for intervention in that there has been accumulating reports on the association between intervention to increase social engagements and health [22].

In the current study, we aimed to examine the risk factors and characteristics of stigmatization among the general population during the COVID-19 pandemic. We assessed various aspects of stigmatization, and its association with social engagements (social capital, social network diversity, and social network size).

## 2. Materials and Methods

### 2.1. Participants

The current study was embedded in the Utsunomiya COVID-19 seroprevalence neighborhood association (U-CORONA) study conducted in Utsunomiya City, Japan [23]. Utsunomiya City is a rural city located in Tochigi Prefecture, Greater Tokyo (1245 people/km^2^ vs. 6449 people/km^2^ in Tokyo), and the majority of residents are older population (% of people aged ≥ 65 years: 25.0% vs. 22.1% in Tokyo). The first survey was conducted from 14 June 2020 to 5 July 2020. The second wave of the survey was conducted between 15 October 2020 and 25 October 2020 (after the second wave of the COVID-19 pandemic in Japan), of which the data were analyzed in the current study. Invitations with a questionnaire were sent to 2290 people in 1000 households randomly selected from Utsunomiya City’s basic resident registry. Among them, 500 people participated in the second wave, returning a valid response to the questionnaire with written informed consent (response rate: 21.8%). We excluded those under the age of 18 years (n = 66) and those without any response to items on stigma (n = 5) and ultimately obtained 429 participants as an analytical sample (see Figure 1 for sampling flowchart). This study was approved by the research ethics committee at Tokyo Medical and Dental University.

### 2.2. Measurements

#### 2.2.1. Stigma

The levels of stigma was measured with a 30-item questionnaire, which was originally developed by public health experts of the current study group (TF, NN, YK, YY, HN) with reference to the 13-item HIV Stigma Scale [24]. The participants were asked about their feelings and thoughts if they or their neighbors were infected with COVID-19 by responding to 15 questions, respectively. Their responses were measured on a four-point Likert scale (1 = strongly disagree to 4 = strongly agree) for each item. The contents of the questionnaire and mean response score for each item are shown in Appendix A. The total stigma score was obtained by calculating the mean score of items and multiplying the total number of items to take into account the missing information (missing in response on 30 stigma items ranged from 1 (7.5%) to 15 (0.2%)). Therefore, the higher scores indicate a stronger stigma. The Cronbach’s alpha for the total stigma score of the current participants was 0.78. The distribution of the total score is shown in Appendix A.

#### 2.2.2. Social Engagements

Community social capital, social network diversity, and social network size were assessed as social engagements. Community social capital was measured with the question “Do you agree or disagree with the following statements? (1) people in your community can be trusted (social trust); (2) this community is close-knit (social tie); (3) people in your community are willing to help their neighbors (mutual aid)” [25,26]. The participant’s responses to the statements were measured with a five-point Likert scale (1 = strongly agree to 5 = strongly disagree). To show stronger social capital with higher scores, we calculated the inversed total score of responses to the three questions as a community social capital score. The Cronbach’s alpha for the community social capital score of the current participants was 0.92.

Social network diversity was evaluated with the number of social roles that participants engaged in on a regular basis after the second wave of the pandemic (after July 2020), according to the following nine roles: spouse, child, parent, relative, neighbor, colleague, member (e.g., club, gym, lesson, religious organizations), friend, and other, based on the Cohen’s Social Network Index [27,28]. A higher score indicates a more diverse social network.

Social network size was evaluated with an open-ended question asking the number of people that participants regularly met and talked to in the aftermath of the second wave of the pandemic (after July 2020).

#### 2.2.3. Demographics

Participants’ sex (male or female), age, education level (junior or high school, vocational school or university, or graduate school), household income (JPY 0 to 3 million, 3 to 6 million, 6 to 10 million, 10+ million; JPY 1 million equaled to USD 9500 in October 2020), the number of comorbidities (i.e., seasonal allergies; asthma or other respiratory diseases; heart diseases; kidney diseases; immune diseases; diabetes or hyperglycemia; malignant tumor; arthritis; frequent and severe headaches; seizure disorders; gastrointestinal disorders; severe acne and other skin diseases; mental illnesses; alcohol or drug use disorders; intellectual disability; autism spectrum disorder; learning disability; tuberculosis; collapsed into 0, 1, and 2+), and depressive symptoms by the Kessler Psychological Distress Scale (K6) score with a validated cut-off (0–4, 5–12, or 13+) [29,30], were assessed. (The Cronbach’s alpha for K6 was 0.90.)

### 2.3. Analysis

First, the correlations between demographic factors and stigma scores were analyzed. The mean score of each stigma item was compared with *t*-test between males and females, and between employment age (18 to 64 years old) and older (≥65 years old). To consider multiple comparisons, *p*-values were corrected with Bonferroni’s method [31]. Next, the total stigma score was compared across demographic factors with *t*-test for sex and age, and with analysis of variance (ANOVA) for education level, household income, the number of comorbidities, and depressive symptoms.

Next, following the previous study [24], we performed principal component analysis on a stigma scale based on a correlation matrix with Varimax rotation. Then, total stigma score and the three factors consisting of stigma scales that were obtained by principal component analysis were regressed over community social capital, social network diversity, and social network size to see the association between social engagements and stigma. The analyses were performed with generalized estimating equations including the clustering at the household level [32] with a robust variance estimator and assumption of unstructured covariance within household, using R package “gee” [33]. The models were adjusted for possible risk factors, i.e., sex, age, education level, household income, comorbidities, and depressive symptoms. All analyses were performed with R version 4.0.5 (R Foundation for Statistical Computing, Vienna, Austria) [34].

## 3. Results

Among 429 participants, 67.4% were employed (mean age 55, interquartile range 40–67), and 46.8% were male. Education level household income and the number of comorbidities were distributed relatively evenly across categories. Four percent of the participants showed clinically severe depressive symptoms. The further details of demographics are shown in Table 1.

The mean score of each stigma scale item is described in Appendix A. Overall, people strongly fear infecting others (mean score: 3.6, SD: 0.74) and causing trouble to their family (mean score: 3.5, SD: 0.73) if they were infected. Additionally, most people agreed with the statement that they would like to “stay away from those infected for a while” (mean score: 3.0, SD: 0.81). The principal component analysis identified the three factors in the stigma scale by examining the scree plot that explains 40% of the variance: social punishment (17% of the total variance was explained), self-deserved (13%), and fear of being infected (10%). The loadings of each component are shown in Appendix A. Briefly, social punishment included the items such as “I will lose my friends if I got infected”, “people who were infected have to move out”, and “I want to avoid future relations with those who are infected”; self-deserve included the items such as “if people in the community got infected, it is a result of their action” and “if people in the community got infected, it is their responsibility”, and fear of being infected included “may infect others” and “may cause trouble for family and colleagues”. The reliability of each component was confirmed by calculating the Cronbach’s alpha of the items of which loadings exceeded 0.40 (social punishment: 0.85, self-deserved: 0.86, fear of being infected: 0.77).

The mean score of each item by sex and age is presented in Figure 2. Overall, females showed higher stigma than males, especially for items on fear of infecting others (*p* < 0.001), desire to hide infection (*p* < 0.01), pressure to move (*p* < 0.05), and obligation to refrain from going out (*p* < 0.001). The difference between the employment age and the older population was not large. Those employed possessed stronger stigmatized views including the desire to hide infection (*p* < 0.05) and indifference towards others being infected (*p* < 0.05) while the older population felt annoyance more if their neighbors were infected (*p* < 0.05). The total stigma score varies by sex (females > males; *p* < 0.05), education level (vocational > university or graduate; *p* < 0.01), and depressive symptoms (5 to 12 > 0 to 4; *p* < 0.01), but not by age, household income, and the number of past diseases (Figure 3 and Appendix A).

Table 2 shows the analysis of the association between social engagements and total and each component of the stigma score. We found no association between social network diversity and size and total stigma score (diversity: B = 0.65, 95% CI = −0.26 to 1.55; size: B = 0.01, 95% CI = −0.04 to 0.05). However, participants with stronger community social capital reported weaker total stigma (B = −0.69, 95% CI = −1.23 to −0.16), mostly driven by the association with the social punishment factor (B = −0.08, 95% CI = −0.12 to −0.03). Interestingly, participants with a more diverse social network reported higher stigmatized views on the self-deserve factor (B = 0.07, 95% CI = −0.0004 to 0.15) and fear of infection factor (B = 0.11, 95% CI = 0.04 to 0.19). Social network size was not associated with any factors (social punishment: B = 0.0001, 95% CI = −0.003 to 0.004; self-deserved: B = −0.001, 95% CI = −0.005 to 0.002; fear of infected: B = 0.003, 95% CI = −0.001 to 0.01).

## 4. Discussion

We showed how the views of the general population could be stigmatized according to the situation if they or others in their community were infected with COVID-19. There were some differences in the level of stigma by demographic factors. The total stigmatization score was higher in females than in males; in those who attained a lower level of education than in those with a higher level of education, and in those who were depressed than in those who were not. Importantly, stigmatization was associated with community social capital; people with stronger community social capital reported less stigmatized views. This association was mostly driven by the association with social punishment factors including fear of losing friends and being excluded from the community. Surprisingly, people with more diverse social networks thought that being infected was self-deserved and felt fear of infection more strongly.

### 4.1. Demographic Factors and Stigmatization

Our findings revealed that females, those who attained an education level of vocational school, and those who are psychologically stressed out (K6 score: 5–12) showed more stigmatized views. Previous studies of stigmatization during the COVID-19 pandemic showed mixed findings on sex differences, with some reporting stronger stigmatization among males than in females [15,17,18], while others showed no differences between males and females [16,35]. More serious stigmatization among females could be attributable to their vulnerable mental health. Females are known to have a higher risk of mental health than males [36], and worse mental health including depressive symptoms [18] and anxiety [16] were associated with more stigmatized views, as shown in the current study as well. The link between depression and stigmatization was suggestive of transmission of depressive symptoms via stigmatizing behaviors during the COVID-19 pandemic, since being stigmatized leads to depressive symptoms [6]. Therefore, targeting individuals who are depressed not only is helpful to them but also has the potential to eliminate the chain of depression transmission.

The relationship between education level and stigmatization is not consistent in the literature. Some studies reported that highly educated people felt/perceived stronger stigmatization, supposedly due to higher sensitivity to social cues (social salience) and worse mental health [16,18]. On the other hand, others reported stronger stigmatization among less educated people [15], which could be attributable to limited available resources, (e.g., income, social support) and engagement in more risky occupations (i.e., roles that require more face-to-face communication). Inconsistency could be explained by the differences in the type of stigma assessed in the studies. A previous study showed a differential association between education and types of stigmas; higher education and stronger enacted stigma featured with prejudice and discrimination were associated, while lower education and stronger internalized stigma such as shame, guilt, and worthlessness were also associated [17]. This accorded with our findings that the stronger stigmatization among those with vocational-level education compared with those with university/graduate school-level education, mostly appeared in the items on the situation where participants themselves were infected (internalized stigma). Our stigma scale did not cover stigma actually experienced due to infection (enacted stigma). Thus, we could not confirm whether enacted stigma was stronger in those with higher education. To elucidate the mechanism across different demographic factors, future studies need to explore the various types of stigmas among the general population.

### 4.2. Social Engagements and Stigmatization

An inverse association between community social capital and stigmatization was identified in our study. We further recognized that this mainly resulted from the social punishment factor. Social capital was repeatedly reported to be beneficial for health [37] but its mechanism is not clear. It can be hypothesized that higher community social capital may increase the dissemination of accurate information and knowledge related to COVID-19 [38], which in turn reduces stigma. Furthermore, a previous study has shown the association between higher health literacy and lower stigmatization [15]. Stronger social capital may also create a cohesive community [38], where everyone belongs to “in-groups” and is not stigmatized as “out-groups”.

On the contrary, larger social network diversity was associated with a stronger perception that infection is self-deserved and scary. Notably, risk perception was tied with stigma [39]. People with diverse social networks would be at a higher risk of being infected than those who interact with a limited number of people; therefore, they may have a stronger risk perception, leading to higher stigmatization, represented by increased fear of being infected. Interestingly, people with a diverse social network may feel being infected is none of others’ business; they are indifferent to both the risk of infecting others and of being infected (a self-deserved factor). Since the current study was cross-sectional, the association could also be explained by a bi-directional association, that is, those who do not feel fear of being infected and who possess individualistic views, (i.e., infection was self-deserved with no relation to others) may continue to have the same social activities as prior to the COVID-19 pandemic. To clarify its directionality, future longitudinal studies are warranted.

### 4.3. Limitations

There are several limitations in the current study. First, this is a cross-sectional study, thus inference of causality is difficult. Second, the survey took place in Utsunomiya City, a largely rural area where older people account for the bulk of the population. Considering the differences in social characteristics between urban and rural areas, the extrapolation of the current findings needs caution. Third, some factors that have been reported to be associated with stigmatization, including the history of COVID-19 infection of a family member and themselves, were not explored here. However, in the current sample, there were only three people who were identified as COVID-19 positive with the chemiluminescence immunoassay (CLIA) method (Shenzhen YHLO Biotech Co., Ltd., Shenzhen, China) [40]; all of them were not recognized as infected prior to the survey. Furthermore, there were no participants whose family members were infected, who had COVID-19-like symptoms, and who potentially came into contact with infected individuals. Thus, we confirmed that in areas with a relatively small infection rate such as Japan, the role of infected individuals in stigmatization would not be as clear as in previous studies conducted in China and India [17,18]. Fourth, all variables assessed here were dependent on self-reporting, where common method bias might be likely. Finally, stigmatization was measured with an original questionnaire. Although it was developed by diverse experts with the consideration of the current pandemic, future studies need to assess the validity and reliability in larger and more diverse samples.

## 5. Conclusions

Enhancing community social capital might be beneficial to reduce stigmatization, especially by reducing social punishment on infected individuals. To reduce stigmatization, increasing health literacy by education would be an option, although its effect would differ according to the education level of the target groups. Moreover, previous randomized controlled trials have revealed the effect of video-based intervention to be short-term [19]. While enhancing social capital is not an easy task, it would be worthwhile delving into along with other potential interventions.

## Figures and Tables

**Figure 1 ijerph-19-09050-f001:**
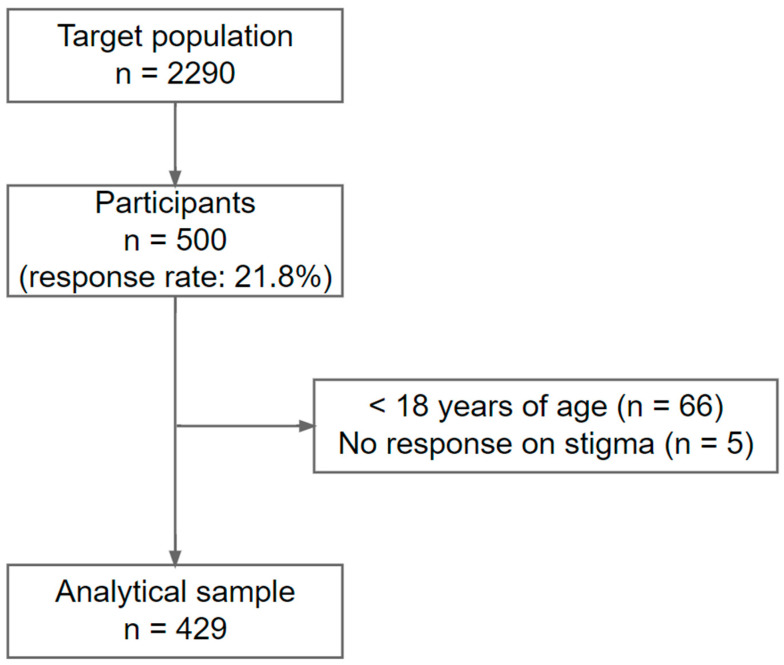
Sampling flowchart.

**Figure 2 ijerph-19-09050-f002:**
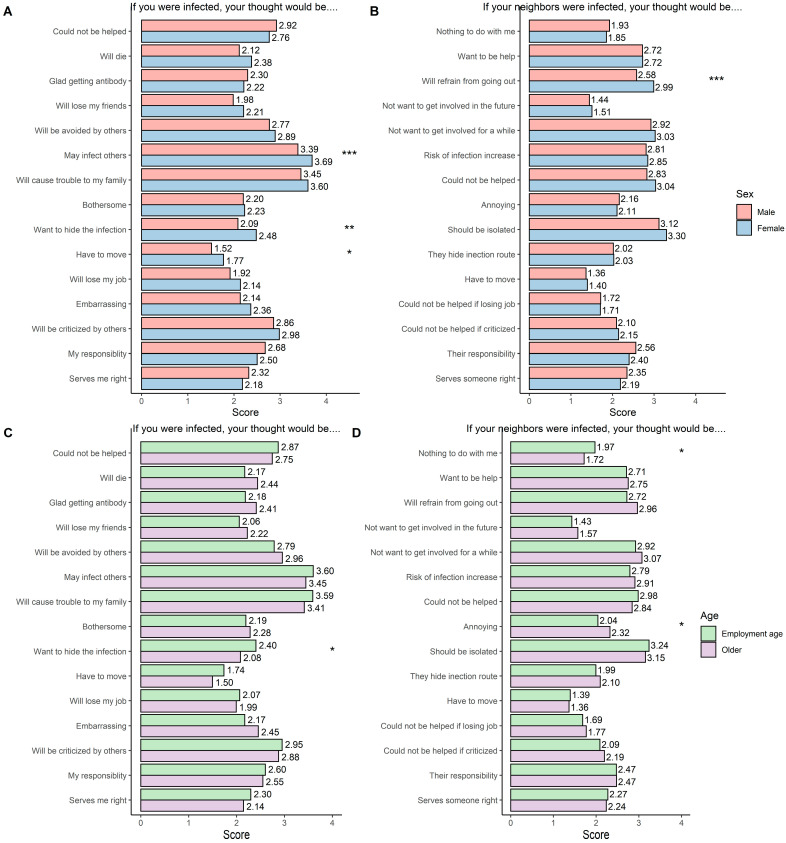
Mean score on each stigma item stratified by sex and age. Mean scores on each COVID-19-related stigma question are shown for males and females (**A**,**B**) and for employment age and older populations (**C**,**D**), separately. Participants were asked if they were infected with COVID-19, how do they think (**A**,**C**), if the neighbors in their community were infected with COVID-19, and how do you think (**B**,**D**). The four-point Likert scale (1 = strongly disagree to 4 = strongly agree) was utilized. Differences in mean scores between males and females, and between employment age and older populations were tested with *t*-test, and *p*-values are shown. *p*-values were adjusted for multiple comparisons with the Bonferroni correction. * Indicates *p* < 0.05, ** indicates *p* < 0.01, and *** indicates *p* < 0.001.

**Figure 3 ijerph-19-09050-f003:**
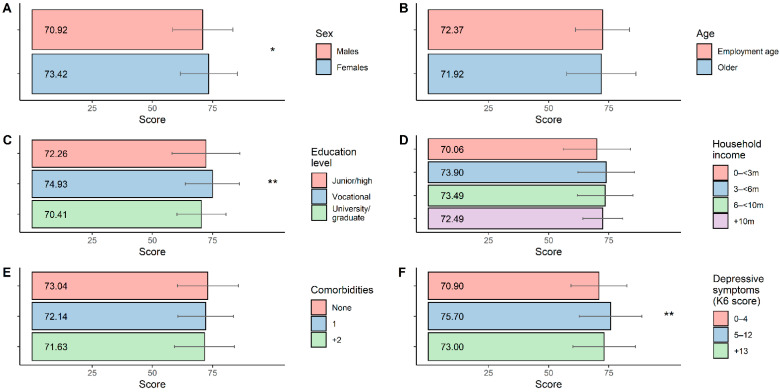
Correlations between participant demographic factors and total stigma scores. Total scores for stigma to both self-infection and others infected (sum of stigma to self-infection and stigma to others infected) are shown. Bars denote mean scores, and upper and lower limits of error bars show mean + 1 SD and mean − 1 SD, respectively. Correlations with participant demographic factors: (**A**) sex, (**B**) age, (**C**) education level, (**D**) household income, (**E**) comorbidities, (**F**) depressive symptoms, were assessed with *t*-test (**A**,**B**) and analysis of variance (ANOVA) (**C**–**F**). * Indicates *p* < 0.05, and ** indicates *p* < 0.01.

**Table 1 ijerph-19-09050-t001:** Sample characteristics (n = 429).

		N (%)
Age	Total (median, IQR)	55 (40, 67)
	Missing	4
	18–65 years old	289 (67.4%)
	>65 years old	140 (32.6%)
Sex	Male	199 (46.8%)
	Female	226 (53.2%)
	Missing	4
Education level	Junior/high	180 (43.2%)
	Vocational	90 (21.6%)
	University/graduate	147 (35.3%)
	Missing	12
Household income (JPY)	0–<3 M	100 (26.4%)
	3–<6 M	122 (32.2%)
	6–<10 M	111 (29.3%)
	+10 M	46 (12.1%)
	Missing	50
Number of comorbidities	None	125 (29.1%)
	1	170 (39.6%)
	+2	134 (31.2%)
Depressive symptoms (K6 score)	0–4	299 (70.2%)
	5–12	110 (25.8%)
	+13	17 (4.0%)
	Missing	3

Abbreviations: JPY, Japanese yen.

**Table 2 ijerph-19-09050-t002:** Associations between social engagements and stigma scores.

	Total Score			Components of Stigma								
				Social Punishment			Self-Deserved			Fear of Infected		
	B	95% CI	*p*-Value	B	95% CI	*p*-Value	B	95% CI	*p*-Value	B	95% CI	*p*-Value
Community social capital	−0.69	−1.23 to −0.16	0.01	−0.08	−0.12 to −0.03	<0.01	−0.02	−0.07 to 0.02	0.29	0.01	−0.04 to 0.05	0.76
Social network diversity	0.65	−0.26 to 1.55	0.16	−0.05	−0.12 to 0.03	0.20	0.07	−0.0004 to 0.15	0.05	0.11	0.04 to 0.19	<0.01
Social network size	0.01	−0.04 to 0.05	0.62	0.00	−0.003 to 0.004	0.97	−0.00	−0.005 to 0.002	0.44	0.00	−0.001 to 0.01	0.12

Adjusted for sex, age, education level, household income, comorbidities, and depressive symptoms. Generalized estimating equations were used.

## Data Availability

The datasets analyzed during the current study are not publicly available since it is the part of a population-based study, conducted by the corresponding author, but are available from the corresponding author upon reasonable request.

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
