# Peer review of "Association between Social Engagements and Stigmatization of COVID-19 Infection among Community Population in Japan"

_ijerph, 2022, doi:10.3390/ijerph19159050_

Round 1
Reviewer 1 Report
Nice work on the figures and the manuscript. Please see file for comments and suggestions to the manuscript. Hope my notes are clear to follow.

Author Response
Interesting article and I do also agree that COVID-19 influenced many aspects of life in the general population. As I reviewed/read the manuscript, I did not find too many English language or grammar issues. The one’s that I have listed under each section are minor, but I hope that the authors will consider making the suggested corrections.
Response:
Thank you for your appreciation on our work and your suggestions to make sentences easier to understand. We handled your comments accordingly.
Line #17 – please consider replacing “danger” or “risks” or “harm”; it fits better.
Response:
We replaced “danger” to “risks”.
“In the face of unknown risks, including the coronavirus disease 2019 (COVID-19) pandemic, we tend to have stigmatized perception.”
Line #26 and #27 – use pleural for males and females. Add “s” to the end. Also, can the depressive symptoms just be depression. When you are depressed, then you present with the symptoms???
Response:
We added the “s” to the end of male and female.
“Overall, females reported higher stigmatized perception of people with COVID-19 than males.”
About the depressive symptoms, we distinguished depression and depressive symptoms as the former indicates medical conditions that doctor diagnoses while the latter is a screened symptoms using self-reported questionnaire.
Line #29 – consider using “comorbidities” instead of “number of past medical diseases”. “Household income and comorbidities were not”.
Response:
We replaced “the number of past medical diseases” to “comorbidities”.
“Lower education and depressive symptoms were also positively associated with higher stigmatization, while age, household income, and comorbidities were not.”
Line #42 – consider changing “change the allocation” to “impact the allocation”.
Line #43 – “exacerbate” makes me think about disease exacerbation, so my choose “intensify” or “amplify”.
Response:
We replaced “change” to “impact”. Also, we replaced “exacerbate” to “amplify”.
“Furthermore, stigmatization could impact the allocation of resources and healthcare access [3,7,8], which may amplify the disparity.”
Line #49 – not sure if “would” is necessary here. Just say put people at high risk …
Response:
We removed “would” from the sentence.
“The coronavirus disease 2019 (COVID-19) pandemic, of which prevention, symptoms, and treatment remained much unknown in 2020 [9], put people at high risk of stigmatization.”
Line #50 – delete “already” and just say we observed ….
Response:
We removed “already” from the sentence.
“We observed stigmatization, especially toward healthcare workers and survivors in the previous pandemic of Middle East Respiratory Syndrome (MERS) and Severe Acute Respiratory Syndrome (SARS), acquired immunodeficiency syndrome (AIDS) and Ebola [10–13].”
Line #56 – change “much” to “extensive” reports ….
Response:
We changed “much” to “extensive”.
“Under the current COVID-19 pandemic, there have been extensive reports on the stigmatization towards healthcare workers [3,6,13,14].”
Line #57 – not sure if “in society” is needed, just say minority groups and you are defining them shortly after.
Line #58 – just change to “those who have a history of incarceration and psychiatric illness”, older population, Asian ethnicity, and those who recovered from COVID-19 are also ….
Response:
We removed in society from the sentence since minorities are explained later as you pointed out, and the sentence was modified as you suggested.
“Minority groups, such as those who have a history of incarceration and psychiatric illness [7], older population [8], Asian ethnicity [3,14–16], and those who recovered from COVID-19 [17,18], are also the targets of stigmatization.”
Line #61 & Line #62 – In addition, people who were residents of highly pandemic areas and survived the disease, felt stigmatized based on their affiliation and beliefs of being excluded. Not sure if I like my wording either but tried.
Response:
Referring your suggestion, we tried to reword the sentence as follows.
“In addition, people who survived the disease and who resided in highly pandemic areas felt stigmatized derived from guilty and shame due to their affiliations and beliefs of being excluded [16–18].”
Line #63 – just a minor suggestion, I think “despite the growing evidence” vs. “despite of the accumulating evidence” fits better.
Line #64 – same minor suggestion, change “limitedly” to “insufficiently” or “inadequately”.
Response:
We changed “despite of the accumulating findings” to “despite the growing evidence”. Also, we changed “limitedly” to “insufficiently”.
“Despite the growing evidence on the people at risk of being stigmatized, there has been little research on who is at risk of stigmatization among the general population, with stigmatization being insufficiently assessed with one or few questions [15,19,20].”
Line #69 – Line #72 – here is an idea for better terminology and flow: those who are highly engaged in their community consider people in their community as in-groups and lower stigmatized perception; whereas those who are not engaged as much would have stronger stigmatization toward people in their community.
Response:
The sentences were replaced as you suggested.
“Those who are highly engaged in their community consider people in their community as in-groups and lower stigmatized perception; whereas those who are not engaged as much would have stronger stigmatization toward people in their community.”
Line #78 – consider changing “aimed to reveal” to “aimed to examine” since this terminology has been used multiple times later in the manuscript. Also, in the last paragraph you have used “stigmatization” four times, may consider eliminating one or two. You may say: In current study, we aimed to examine the risk factors and characteristics of stigmatization among the general population during COVID-19 pandemic. We assessed various aspects of stigmatization …. (just eliminated one).
Response:
As you suggested the sentence was modified to avoid redundant use of stigmatization.
“In the current study, we aimed to examine the risk factors and characteristics of stigmatization among the general population during the COVID-19 pandemic. We assessed various aspects of stigmatization, and its association with social engagements (social capital, social network diversity, and social network size).”
Line #85 – you listed current study, should there be a time frame since you go on to report “the second wave was conducted between October 15 – 25, 2020.
Response:
We excluded the date of the first survey to avoid misunderstanding since we only used the result of the second wave. However, upon your suggestion, we added for the first wave as well.
“The first survey was conducted from 14 June 2020 to 5 July 2020. The second wave of the survey was conducted between 15 October 2020 and 25 October 2020 (after the second wave of the COVID-19 pandemic in Japan), of which the data were analyzed in the current study.”
Line #93 – may use <18 years of age (n=66). You have it listed this way in the figure.
Response:
As you suggested, the figure 1 was modified.
Line #101 – delete the second stigma before the questionnaire, you say it at the beginning of the sentence.
Response:
The second “stigma” was deleted.
“The levels of stigma was measured with a 30-item questionnaire, which was originally developed by public health experts of the current study group (TF, NN, YK, YY, HN) with reference to the 13-item HIV Stigma Scale [24].”
Line #103 and Line #104 – delete “would-be” then revise the sentence to: the participants were asked about their feelings and thoughts if they or their neighbors were infected with COVID-19 by responding to 15-questions respectively. Their responses were measured on a 4-point Likert scale (….). end here
Response:
The sentences were changed as you suggested.
“The participants were asked about their feelings and thoughts if they or their neighbors were infected with COVID-19 by responding to 15-questions, respectively. Their responses were measured on a four-point Likert scale (1 = strongly disagree to 4 = strongly agree) for each item.”
Line #109 - not sure if “missingness” is appropriate, may just say for missing data or information.
Response:
We changed “missingness” into “missing information”.
“The total stigma score was obtained by calculating the mean score of items and multiplying the total number of items to take into account the missing information (missing in response on 30 stigma items were ranged from 1 (7.5%) to 15 (0.2%)).”
Line #118 – here you can also say, the participants response to the statements was measured with a 5- point …….end at the end of).
Response:
The sentence was modified as suggested.
“The participants’ responses to the statements were measured with a five-point Likert scale (1 = strongly agree to 5 = strongly disagree).”
Line #123 – “participants engaged”! since you are also saying “evaluated”
Line #125 – delete “group”, not necessary. Also, either use i.e., or e.g., not very many listed.
Response:
The term was changed to past tense and “group” was deleted.
“Social network diversity was evaluated with the number of social roles that participants engaged in on a regular basis after the second wave of the pandemic (after July 2020), according to the following nine roles: spouse, child, parent, relative, neighbor, colleague, member (e.g., club, gym, lesson, religious organizations), friend, and other, based on the Cohen’s Social Network Index [27,28].”
Line #129 – same as above “met and talked to”
Response:
The terms were changed to past tense.
“Social network size was evaluated with an open-ended question asking the number of people that participants regularly met and talked to in the aftermath of the second wave of the pandemic (after July 2020).”
Line #132 – consider changing “educational attainment” to just “education level”.
Response:
“Educational attainment” was changed to “education level” throughout the manuscript.
Line #133 – what this amount to per US dollar??? Also, need a space between Y and 0.
Line #134 – as previously suggested, “past medical diseases” to “comorbidities”. Also, no need to say counting the number of following PMH. Just say number of comorbidities (i.e., seasonal allergies, asthma or other respiratory disorder, heart disease, DM, kidney disease, etc.); not sure if you need to list them all, just few big examples.
Line #138 – consider changing stomach and duodenum to gastrointestinal. Line #139 – the new terminology is “alcohol or drug use disorder”.
Line #140 and #141 – sounds like there is something missing. In this paragraph, are these the items measured in the demographic section? Why depression is not considered as comorbidity as well? Also, I think you should consider using the full term for Kessler Psychological Distress (K6) questionnaire score. I was for sure not familiar with this scale!! Excited to learn something new.
Response:
The terms in the sentence were modified as you suggested and further explanation was added where needed.
About the comorbidities, we asked whether they were diagnosed as depression or anxiety, which was expressed as “mental illnesses”.
“Participants’ sex (male or female), age, education level (junior or high school, vocational school or university, or graduate school), household income (JPY 0 to 3 million, 3 to 6 million, 6 to 10 million, 10+ million; JPY 1 million equaled to USD 9,500 in October 2020), the number of comorbidities (i.e., seasonal allergies; asthma or other respiratory diseases; heart diseases; kidney diseases; immune diseases; diabetes or hyperglycemia; malignant tumor; arthritis; frequent and severe headaches; seizure disorders; gastrointestinal disorders; severe acne and other skin diseases; mental illnesses; alcohol or drug use disorders; intellectual disability; autism spectrum disorder; learning disability; tuberculosis; collapsed into 0, 1, and 2+), and depressive symptoms by the Kessler Psychological Distress Scale (K6) score with a validated cut-off (0-4, 5-12, or 13+) [29,30], were assessed.”
Line #144 – as mentioned above; males and females.
Line #145 – a minor comment to change “equal or above 65 years old” to “≥ 65 years of age”. This format is accepted. Also, maybe using “employment age” for “between working age”.
Response:
The expressions were changed as suggested.
“The mean score of each stigma item was compared with t-test between males and females, and between employment age (18 to 64 years old) and older (≥ 65 years old).”
Line #148 – change “attainment” to “education level” as suggested previously for consistency.
Line #149 – same with “number of past medical diseases” to “number of comorbidities”.
Response:
The words were changed as proposed.
“Next, the total stigma score was compared across demographic factors with t-test for sex and age, and with analysis of variance (ANOVA) for education level, household income, the number of comorbidities, and depressive symptoms.”
Line #151 – Line #153 – you lost me here; hope I’m not the only one. Not sure what we are trying to say. ..and the total and each component of stigma???
Response:
We explained the analysis more precisely here as follows.
“Then, total stigma score and the three factors consisting stigma scale that were obtained by principal component analysis were regressed over community social capital, social network diversity, and social network size to see the association between social engagements and stigma. The analyses were done with generalized estimating equations including the clustering at the household level [32] with a robust variance estimator and assumption of unstructured covariance within household, using R package “gee” [33].”
Line #162 – Consider changing “working-age population’ to “67.4% were employed”.
Response:
The sentence was changed as suggested.
“Among 429 participants, 67.4% were employed (mean age 55, interquartile range 40-67), and 46.8% were male.”
Line #163 – Line #164 – revise to “education level, household income, and comorbidities were….
Response:
The terms were corrected as previously suggested.
“Education level household income, and the number of comorbidities were distributed relatively evenly across categories.”
Line #170 – should “trouble” be “harm”?
Response:
We chose “trouble” over “harm” since the concern assessed here is not about infecting family but about causing trouble by having infected person within family member (e.g., need to take a day off, need to be taken care of).
Line #189 – delete the “s” in population. Also, change to those employed, possessed …..
Line #191 – delete the second “more” after annoyance.
Response:
“populations” changed to “population” and “working age population” changed to “those employed”. Also, the second “more” was deleted.
“The difference between employment age and older population was not large. Those employed possessed stronger stigmatized views including desire to hide infection (p < 0.05) and indifference towards others being infected (p < 0.05) while the older population felt annoyance more if their neighbors were infected (p < 0.05).”
Line #224 – I believe this should be Table #2 since your first table is on Line #167. Line #213 you have listed as Table 2 shows the analysis, is this the same table as Line #224? Please use the correct title for this table (2).
Response:
Thank you for pointing out. We changed the title of Table 2 (shown in Line #224).
Line #229 – consider replacing “showed” with “revealed” or “discovered” or “explored”.
Line #230 – I think “of” after situation should be deleted. May also consider rewording the sentence “if they or others in their community were infected with COVID-19”. Combine since the same concept.
Response:
We deleted “of” after “the situation” and combined the two sentences.
“We showed how the views of the general population could be stigmatized according to the situation if they or others in their community were infected with COVID-19.”
Line #231 – delete “s” in levels.
Response:
“s” was deleted.
“There were some differences in the level of stigma by demographic factors.”
Line #246 – not sure “severer” is appropriate terminology, may revise to just “severe stigmatization”…
Response:
We would like to make comparison here, thus modified as “more serious stigmatization”.
“More serious stigmatization among females could be attributable to their vulnerable mental health.”
Line #248 – not sure if you have to say depressive symptoms, just say “depression”.
Response:
We would like to discuss as same as in Line #26 and #27 that we distinguished depression and depressive symptoms as the former indicates medical conditions that doctor diagnoses while the latter is a screened symptoms using self-reported questionnaire. However, if you think “depression” is more appropriate, we are welcome to change.
Line #252 – Line #253 – Therefore, targeting individuals who are depressed not only is helpful to them but has the potential to eliminate the burden to their loved ones.
Response:
We would like to discuss about transmission of depressive symptoms generally, and not limited to transmission between partners or family members (as suggested with “their loved one”). Therefore, we modified the sentence here as follows.
“Therefore, targeting individuals who are depressed not only is helpful to them but also has the potential to eliminate the chain of depression transmission.”
Line #255 – as mentioned above, change “educational attainment” to “education level”.
Response:
“Educational attainment” was changed to “education level”.
“The relationship between education level and stigmatization is not consistent in the literature.”
Line #256 – revise to: some studies reported that highly educated people felt/perceived stronger stigmatization”.
Response:
We modified as suggested.
“Some studies reported that highly educated people felt/perceived stronger stigmatization, supposedly due to higher sensitivity to social cues (social salience) and worse mental health [16,18].”
Line #259 – change the second less to “limited available resources”.
Response:
We changed “less” to “limited”.
“On the other hand, others reported stronger stigmatization among less educated people [15], which could be attributable to limited available resources (e.g., income, social support) and engagement in more risky occupations (i.e., roles that require more face-to-face communication).”
Line #262 – Line #266 – it may just be me; I am having hard time following this section.
Response:
We avoided to use brackets here and explained more simply as follows.
“A previous study showed differential association between education and types of stigmas; higher education and stronger enacted stigma featured with prejudice and discrimination were associated, while lower education and stronger internalized stigma such as shame, guilt, and worthlessness were also associated [17].”
Line #272 – just end the sentence with “higher education”.
Response:
The sentence was modified.
“Thus, we could not confirm whether enacted stigma was stronger in those with higher education.”
Line #276- Line #277- change “found” to “identified”. And also, “showed” to “recognized”, this is to limit the number of times “showed” has been repeated. Or, you can reverse the sentence to: We identified and inverse association between community social capital and stigmatization in our study. Then, we discovered that this mainly…..
Line #278 – here you can use “reported” for “shown” it makes the sentence flow better.
Response:
We changed “found” to “identified”, “showed” to “recognized”, and “shown” to “reported”.
“An inverse association between community social capital and stigmatization was identified in our study. We further recognized that this mainly resulted from the social punishment factor in stigma. Social capital was repeatedly reported to be beneficial for health [37] but its mechanism is not clear.”
Line #280 – Change “diffuse” to “dissemination” and “correct information” to “accurate information” or “valid information” for better terminology.
Response:
We changed the wordings as suggested.
“It can be hypothesized that higher community social capital may increase the dissemination of accurate information and knowledge related to COVID-19 [38], which in turn reduces stigma.”
Line #285 – Line #286 – Not sure if the sentence is complete where you report …infection is self-deserved and fear of infection????
Response:
We clarified the sentence as follows.
“On the contrary, larger social network diversity was associated with stronger perception that infection is self-deserved and scary.”
Line #288 – change “meet” to “interact”.
Response:
We changed “meet” to “interact”.
“People with diverse social network would be at higher risk of being infected than those who interact with limited number of people;”
Line #296 – What do you mean by “to clarify its directionality”?
Response:
In the previous sentence, we discussed about potential bi-directional association between social engagements and stigma. “To clarify its directionality” means to find out whether social capital decrease stigmatization or having less stigmatized perception cause stronger social capital.
Appreciate authors recognizing the various limitations.
Response:
Thank you for your appreciation.
Line #301 – You have largely and large here in one sentence. Consider changing to: a largely rural area where older people account for the bulk/majority of population.
Response:
We avoided to use “large” repeatedly by accepting your correction.
“Second, the survey took place in Utsunomiya City, a largely rural area where older people account for the bulk of population.”
Line #310 – change “showed” to “had COVID-19- like symptoms
Response:
We changed “showed” to “had”.
“Furthermore, there were no participants whose family members were infected, who had COVID-19-like symptoms, and who potentially came into contact with infected individuals.”
Line #313 and #314 - May consider combining these two sentences as all variables assessed here were dependent on self-reporting, where common method bias might be likely.
Response:
The sentences were combined as suggested.
“Fourth, all variables assessed here were dependent on self-reporting, where common method bias might be likely.”
TABLES: Table #1 – change attainment to level, number of past diseases to number of comorbidities; K6 total score to just K6 score
Response:
The terms in table 1 were modified.
Table #2 – Very hard to follow, not sure if it can be adjusted; correct the title and it should be table #2.
Response:
I agree that values in Table 2 were difficult to read. The title was corrected.
Figure 2 – Appreciate the detailed content; A and C all the variables are capitalized, except the “G” in the glad getting antibody, please correct.
Response:
Thank you for your appreciation. We corrected to the capital letter.
Figure #3 - A under sex you have male and then females, make them both pleural; C as suggested education level; E comorbidities vs. past medical diseases.
Response:
The terms were corrected.

Reviewer 2 Report
It is a very interesting study although, as mentioned by the authors, difficult to extrapolate to other national and regional realities. Overall, the methodology is sound and coherent, results are interesting and leave ground for other follow-up studies.
There are some small suggestions that may improve readability:
Materials and Methods:
This section will benefit of a demographic description of the study's area. There is reference to this in the Limitations but it should definitely also be present in the methods section.
Not clear the reference of a second wave of the survey. It seems data was collected exclusively between 15-25 October in one go.
All measurements, especially stigma which was not previously validated, should be accompanied by reliability values (not only for the components of stigma).
Results:
line 191 - delete 'more'
lines 213 - reference to Table 2 without the correct table number and caption in lines 224-225
Author Response
-It is a very interesting study although, as mentioned by the authors, difficult to extrapolate to other national and regional realities. Overall, the methodology is sound and coherent, results are interesting and leave ground for other follow-up studies.
-Response:
Thank you for taking time to review our manuscript. We are pleased that you found this study was interesting and warranted the future studies despite of the limited generalizability.
-This section (Materials and Methods) will benefit of a demographic description of the study's area. There is reference to this in the Limitations but it should definitely also be present in the methods section.
-Response:
We added demographic explanation in the methods section as follows.
“The current study was embedded in the Utsunomiya COVID-19 seROprevalence Neighborhood Association (U-CORONA) study conducted in Utsunomiya City, Japan [23]. Utsunomiya City is a rural city located in Tochigi Prefecture, Greater Tokyo (1,245 people/km2 vs 6,449 people/km2 in Tokyo), and the majority of residents is older population (% of people aged ≥ 65 years: 25.0% vs 22.1% in Tokyo).”
-Not clear the reference of a second wave of the survey. It seems data was collected exclusively between 15-25 October in one go.
-Response:
The details of the survey were described in Y Koyama et al., 2021, Brain Behav Immunity. Briefly, we sent invitations and questionnaires beforehand and collected during the survey, between 15-25 October 2020. We added the date of the first wave of the survey to clarify the survey flow.
“The first survey was conducted from 14 June 2020 to 5 July 2020. The second wave of the survey was conducted between 15 October 2020 and 25 October 2020 (after the second wave of the COVID-19 pandemic in Japan), of which the data were analyzed in the current study.”
-All measurements, especially stigma which was not previously validated, should be accompanied by reliability values (not only for the components of stigma).
-Response:
We added the Cronbach’s alpha for each scale where appropriate.
“Therefore, the higher scores indicate a stronger stigma. The Cronbach’s alpha for the total stigma score of the current participants was 0.78. The distribution of the total score is shown in Supplemental Figure 1.”
“To show stronger social capital with higher scores, we calculated the inversed total score of responses to the three questions as a community social capital score. The Cronbach’s alpha for the community social capital score of the current participants was 0.92.”
“(The Cronbach’s alpha for K6 was 0.90.)”
-line 191 - delete 'more'
-Response:
The second “more” was deleted.
“The difference between employment age and older population was not large. Those employed possessed stronger stigmatized views including desire to hide infection (p < 0.05) and indifference towards others being infected (p < 0.05) while the older population felt annoyance more if their neighbors were infected (p < 0.05).”
-lines 213 - reference to Table 2 without the correct table number and caption in lines 224-225
-Response:
We modified the title of Table 2.
